



# Insignificant but robust decrease of ENSO predictability in an equilibrium warmer climate

Yiyu Zheng[1], Maria Rugenstein[2], Patrick Pieper[1], Goratz Beobide-Arsuaga[1], and Johanna Baehr[1]

[1]Institute of Oceanography, The Center for Earth System Research and Sustainability, Universität Hamburg, Hamburg, Germany

[2]Department of Atmospheric Science, Colorado State University, Fort Collins, CO, USA

**Correspondence:** Yiyu Zheng (Yiyu.Zheng@colostate.edu)

**Abstract.** Responses of El Niño-Southern Oscillation (ENSO) to global warming remain uncertain, which challenges ENSO forecasts in a warming climate. We investigate changes in ENSO characteristics and predictability in idealized simulations with quadrupled $CO_2$ forcing from seven general circulation models. Comparing the warmer climate to control simulations, ENSO variability weakens, with the neutral state lasts longer, while active ENSO states last shorter and skew to favor the La Niña

state. Six-month persistence-assessed ENSO predictability slightly reduces in five models and increases in two models under the warming condition. While the overall changes in ENSO predictability are insignificant, we find significant relationships between changes in predictability and intensity, duration and skewness of the three individual ENSO states. The maximal contribution to changes in the predictability of El Niño, La Niña and neutral states stems from changes in skewness and events' duration. Our findings show that a robust and significant decrease in ENSO characteristics does not imply a similar change

in ENSO predictability in a warmer climate. This could be due to model deficiencies in ENSO dynamics and limitations in persistence model when predicting ENSO.

## 1   Introduction

Improving ENSO predictions is crucial to project large-scale climate variability, which is of great scientific interest and societal need (Timmermann et al., 2018; Christensen et al., 2013; L'Heureux et al., 2020). Tremendous research efforts have improved

the understanding of ENSO variability (e.g., Wang, 2018) and developed reliable ENSO predictions at 6-12 lead months in the present climate (e.g., Cane et al., 1986; Chen et al., 1997; Chapman et al., 2015; Tang et al., 2018; Ham et al., 2019). Anthropogenic climate change could alter ENSO characteristics in the future. Many studies have analyzed the impacts of climate change on ENSO characteristics in different warming scenarios with various models, but the impacts of climate change on ENSO predictability have not been investigated with similar rigor, due to large uncertainties in the simulated ENSO response.

The climate model deficiencies, the short observational record, and the internal variability have prevented us from drawing robust conclusions about likely changes of ENSO characteristics in the future (e.g., Beobide-Arsuaga et al., 2021; Zhu, 2021; Stevenson et al., 2010; Maher et al., 2018; Zheng et al., 2018). Studies have reviewed ENSO response to external $CO_2$ forcing in models from the Couple Model Intercomparison Project phase 3 and 5 (CMIP3 and CMIP5), and the results show that changes in ENSO amplitude and frequency under the forcings are model-dependent (Guilyardi, 2006; Guilyardi et al., 2012).





Recent studies suggest that improved climate simulations reduce the model spread of ENSO responses to climate change. For example, Fredriksen et al. (2020) showed that CMIP6 models simulate ENSO spectra, amplitude and teleconnection in warming scenarios in higher agreement than models in CMIP phase 5 and 3. In addition to model development, longer simulations, especially those longer than 300 years, are shown to reduce uncertainties in ENSO simulation by stabilizing the internal variability (e.g., Sun et al., 2018; An et al., 2008; Knutson et al., 1997). Callahan et al. (2021) used millennial-length

simulations from the Long Run Model Intercomparison Project (LongRunMIP) to show that ENSO amplitude and frequency changes were stronger and more consistent across models when the climate system equilibrated a $CO_2$ forcing than during the transient climate state. Building on this finding, we ask if a robust change in ENSO characteristics will lead to a robust change in ENSO predictability.

ENSO predictability in the present climate is related to ENSO characteristics. Jin et al. (2008) investigated the skill of 6-

month ENSO predictions in the hindcasts of 10 coupled general circulation models (GCMs) and showed that the larger the ENSO intensity, the easier the predictions, and that El Niño and La Niña states are better predictable than the neutral state. Liu et al. (2022) studied retrospective ENSO forecasts in an ensemble prediction system in Community Earth System Model (CESM) and confirmed that stronger ENSO events exhibit higher predictability.

The relationship between ENSO predictability and characteristics has also been investigated in a warmer future. DelSole

et al. (2014) ascribed a decreased seasonal ENSO predictability to a decreased ENSO variance under a fixed 2095 climate forcing in the Community Climate System Model, version 4. Contractively, Berner et al. (2020) showed an increased seasonal to interannual predictability in a strong warming scenario in CESM, which was attributed to an increased ENSO amplitude and frequency. However, previous studies have neither provided a model-intercomparison of ENSO predictability change in a warmer climate, nor an investigation of predictability changes in El Niño, La Niña and neutral states separately.

In this paper, we compare changes in ENSO predictability in a quasi-equilibrated warmer climate across seven models. We analyze ENSO predictions using a persistence model, and we investigate predictability of the time series of the Niño index, and its decomposition into the three ENSO phases. Based on the robust decrease in ENSO characteristics in an equilibrium climate (in 8 out of 10 models as shown in Callahan et al. (2021)) in combination with the inherent relationship between ENSO characteristics and predictability, we could expect that ENSO predictability decreases robustly in a quasi-equilibrated warming

climate, or at the very least that the ENSO characteristics-predictability relationship sustains in such a climate.

This study is organized as follows: in Section 2, we introduce the data and method. In Section 3, we analyze changes in ENSO characteristics and predictability in response to $CO_2$ forcing, and we regress changes in ENSO predictability onto changes in ENSO characteristics to determine the relationship. In section 4, we discuss implications of these findings and summarize the results.

## 2   Data and Method

We use seven atmosphere-ocean general circulation models from the LongRunMIP archive (Rugenstein et al., 2019). We compare their pre-industrial control simulations of constant 1850-forcing (hereinafter referred to as *control*) with their step-



forcing simulations of quadrupled $CO_2$ (hereinafter referred to as *abrupt4xCO2*). This forcing level is chosen because it shows better signal-to-noise ratio than the doubling of $CO_2$, and its simulations are available in most of the LongRunMIP models.

We treat the period between year 150 and the end of the simulation as quasi-equilibrated (Table 1; Callahan et al. (2021)). Our results are not sensitive to this cut of time. In model simulations, we analyze monthly surface air temperature (TAS; Fig. 1) instead of sea surface temperature (see Rugenstein et al. (2019) for data availability). For the observational record, we analyze TAS by using the Godded Institute for Space Studies Surface Temperature Analysis version 4 (GISTEMPv4) (GISTEMP Team, 2022; Lenssen et al., 2019). We compare changes in the relationship between ENSO characteristics and predictability

calculated from the GISTEMPv4 to the one that calculated from *control*. In comparison to the relatively short observational record that shows only 65 El Niño and La Niña events from 1880 to present, we detect over 300 occurrences for both El Niño and La Niña events per model in *control*, accentuating the increased statistical robustness and reliability of our analysis by using millennial-length simulations.

To evaluate ENSO simulations in *control*, we calculate the Niño3.4 index by computing time series of TAS anomalies

averaged over the Niño3.4 region (170°W–120°W, 5°N–5°S) with centered 30-year based periods that update every five years (Lindsey, 2013). The *control* show fair distributions of the Nino3.4 index compared to the observational record (Fig. 2a-g), except for GISSE2R that simulates too many neutral states and too few and weak active ENSO states (Fig. 2d). The power spectra of the Niño3.4 index is model-dependent in *control* in terms of the dominant frequency. Three models agree with the observations that ENSO occurs every 3 to 7 years (Fig. 2i,l,m), three models simulate a more frequent occurrence (Fig.

2h,j,k), and one model simulates a less frequent occurrence (Fig. 2n). To analyze model biases in ENSO dynamics, there are generally two most important atmospheric feedbacks that dominate ENSO dynamics: the negative net heat flux feedback and the positive Bjerknes feedback (e.g., Lloyd et al., 2009). Due to data availability, we only compute the net shortwave heat flux feedback, regressing TAS anomalies in the Niño3.4 region onto the net shortwave heat flux anomalies in the combined Niño3 and Niño4 region (160°E–90°W, 5°N–5°S; Bayr et al. (2019)). Six models simulate the observed negative feedback parameter,

albeit with too low or too high magnitudes (Fig. 3). IPSLCM5A simulates an erroneously positive feedback, indicating that the IPSLCM5A mistakenly simulates ENSO events to be shortwave-driven (Bayr et al., 2019).

To separate ENSO responses to $CO_2$ forcing into the three ENSO states, we define El Niño and La Niña events as exceedance of half a standard deviation of the Niño3.4 index for at least six consecutive months (Fig. 4a-g; Bellenger et al. (2014)). Our results are not sensitive to the choice of different exceedance thresholds. In addition, we define "quasi El Niño" and "quasi La

Niña" events as exceeding the threshold but not lasting long enough to be classified as an active event. We then investigate the time series of the Niño3.4 index (hereinafter referred to as "mean"), as well as neutral, El Niño, and La Niña composites (hereinafter referred to as "states" collectively, and as "active ENSO states" when exempting the neutral state).

For ENSO mean characteristics, we define the ENSO amplitude as the standard deviation of the Niño3.4 index, and the frequency as the area of the Niño3.4 spectrum that lies above the Markov red-noise spectra (Wilks (2011); Fig. 4h-n). For the

characteristics of each ENSO state (excluding quasi-events), we define the events' intensity as the average peak of the Niño3.4 index of the El Niño and La Niña event, and the events' duration as the average count of months of the El Niño, La Niña or





neutral event. To measure the magnitude of ENSO asymmetry, we compute the skewness of the Niño3.4 index distribution (Kohyama et al., 2018).

To compute ENSO predictability in *control* and *abrupt4xCO₂*, we generate a persistence forecast of the Niño3.4 index. This persistence forecast uses a first-order auto-regressive model initialized in November, which produces the highest skill in short lead times (Knaff and Landsea, 1997; Jin et al., 2008) and allows us to investigate the sensitivity of the spring predictability barrier to global warming. We measure the predictive skill by a categorical metric, the Accuracy, which is obtained from a contingency table by calculating the proportion of correct predictions for each ENSO state relative to the sum of all predictions (Wilks, 2011):

$$Accuracy = \frac{(hits + correct\ negatives)}{(hits + correct\ negatives + misses + false\ alarms)}. \tag{1}$$

The hits are predictions for which a state that is forecasted to occur does occur, the correct negatives are the number of states forecasted to not occur and do not occur, the misses are states forecasted to not occur but occur, and the false alarms are states forecasted to occur but do not occur. We calculate the accuracy of ENSO states for the first six lead-months (as shown in Fig. 5), which is hereafter referred to as November to April. Predictability is defined as the area under the skill curves across the six lead months.

Throughout the paper, we focus on the change of ENSO characteristics and predictability in response to a warmer climate. The change is computed by subtracting the averaged *control* from every time step in *abrupt4xCO₂*. We test the significance of changes by one-sided bootstrapping, of which the null hypothesis (changes are not signficant) will be rejected at the 5% significance level. For changes in the mean characteristic, we bootstrap the Niño3.4 index in *control* and *abrupt4xCO₂*; for changes in states' characteristic, we bootstrap the intensities or durations of each ENSO state. For changes in ENSO predictability, we convert contingency tables into arrays, and we bootstrap these arrays at the 5% significance level. For example, we assign four letters to represent each constituent ("a" for hits, "b" for correct negatives, "c" for misses, and "d" for false alarms) and we assume a contingency table in *control* that has one hit (a), two correct negatives (b, b), three misses (c, c, c) and four false alarms (d, d, d, d). We convert the contingency table into an array ([a, b, b, c, c, c, d, d, d, d]) and bootstrap the array with replacement 5000 times. We now obtain 5000 values of accuracy for each *control* and repeat this process for *abrupt4xCO₂*. For each lead month, we repeat this bootstrap procedure and calculate changes in accuracy. At last, we test if our original predictability change is statistical significantly positive or negative.

To investigate relationships between changes in ENSO predictability and characteristics (hereinafter referred to as "ENSO characteristics-predictability relationship"), we employ a linear regression analysis. We break the time series of the Niño3.4 index into running 30-year windows that update every five years, calculate the mean and the states' predictability and characteristics for each 30-year period, and regress the changes in ENSO predictability onto changes in ENSO characteristics. We test whether the regression slopes significantly differ from zero by bootstrapping the dependent and independent variables in the linear equations. In addition, we calculate $R^2$ values to determine the contribution of ENSO characteristic changes to ENSO predictability changes.



## 3 Results

### 3.1 Changes in ENSO characteristics

ENSO weakens and reddens in *abrupt4xCO₂* compared to *control* in most of the models (Fig. 4a-g and 6a,d). Distributions of the Niño3.4 index narrows in six out of seven models and their central peaks intensify (Fig. 4a-g), reflecting a decrease in ENSO amplitude. Six models show significantly decreased ENSO amplitudes in *abrupt4xCO₂* between -0.03 and -0.34 °C (4 to 61% compared to their *control* values; Fig. 6a). They also show a reddening of high frequency in *abrupt4xCO₂* (Fig. 4h-n) accompanied by a significant decrease in the power of high frequencies (-2.35 to -21.0 °C/year, which is 24 to 92% of the values in *control*; Fig. 6d). MPIESM12, being the only exception, shows an increase in ENSO amplitude of 0.15 °C (21%) and an increase in ENSO frequency of 8.53 °C/year (85%).

Both the El Niño and La Niña state becomes weaker and shorter (Fig. 6). The El Niño state significantly decreases its intensity in six out of seven models (-0.14 to -0.54 °C, 14 to 44% relative to *control*; Fig. 6b) and significantly decreases its duration in the same six models (-1.49 to -12.0 months, 11 to 59% relative to *control*; Fig. 6e). Similarly, the La Niña state significantly decreases its intensity in four models (-0.18 to -0.42 °C, 19 to 36% relative to *control*; Fig. 6c) and significantly decreases its duration in six models (-1.16 to -10.3 months, 10 to 52% relative to *control*; Fig. 6f).

The decreased durations of both active states are accompanied by an increased duration of the neutral state. Five models simulate a significantly longer neutral state in *abrupt4xCO₂* (3.73 to 60.2 months, 28 to 60.5% relative to *control*; Fig. 6g).

The difference in duration changes in both active states suggests changes in the ENSO nonlinearity. To exemplify changes of the asymmetry in Niño3.4 distributions, we scrutinize HadCM3L (Fig. 4e). In *control*, the right-side tail of the histogram of HadCM3L extends further than the left-side tail, denoting more extreme El Niño than La Niña events. In addition, the histogram peaks around the threshold that differentiates the neutral from the La Niña state, leading to more quasi La Niña than quasi El Niño events. In *abrupt4xCO₂*, the right tail becomes shorter, while the left tail remains, resulting in fewer extreme El Niño events. Additionally, the peak of the distribution moves to the right, resulting in more quasi El Niño events relative to *control*. Four models simulate a significantly decreased skewness of the Niño3.4 distribution (-0.12 to -1.07; Fig. 6h), following that quasi El Nino increases more than quasi La Nina events, while extreme El Nino decreases more than extreme La Nina events in *abrupt4xCO₂*.

### 3.2 Changes in ENSO predictability

ENSO becomes less predictable in five out of seven models, although the changes are mostly insignificant according to our bootstrap test (Fig. 5a-g and 7a). In *control*, the mean persistence skill starts at around 0.95 in all seven models, denoting that the persistence model is able to correctly predict 95% of ENSO states in November. From December onward, the skill drops to around 0.72 for April, denoting that 72% of ENSO states are correctly predicted after six months. In *abrupt4xCO₂*, all models first show equivalent skills from November to January as in *control*. Then, five models show a decrease in skills, with around 70% of correct predictions of ENSO states for April, while GISSE2R and MPIESM12 show a small increase in skill. We





quantify changes in the area under the skill curve as the change in predictability, with only HadCM3L showing a significant decrease in the mean predictability in *abrupt4xCO₂* (-0.42, 9% relative to *control*; Fig. 7a).

The predictability of the El Niño state decreases similarly as the mean predictability, while the predictability of the La Niña state shows no significant change (Fig. 5h-n and 7b-c). In *control*, the persistence skills of active states reduce from 0.95 to 0.8 from November to April in all seven models, being indistinguishable from the mean persistence skill. In *abrupt4xCO₂*, the skill of predicting El Niño state decreases similarly as in *control* from November to January, then changes sharply and model-dependently form February onward. In comparison, between November and April, the skill of the La Niña state changes less than the skill of the El Niño state. Four models show a decrease in the predictability of the El Niño state, with HadCM3L

being again the only model that exhibits a significant change (-0.54, 11% relative to *control*; Fig. 7b). Five models agree on a decrease in the predictability of the La Niña state, while none of the changes are significant (Fig. 7c).

The neutral state in both *control* and *abrupt4xCO₂* is harder to predict than the mean and the active states, and its change in predictability exhibits the largest magnitude (Fig. 5h-n and 7d). In *control*, the skill of the neutral state begins at 0.9 in November and decreases to around 0.65 in April. In *abrupt4xCO₂*, the skill decreases similarly to *control* in the first three

lead months and changes sharply and model-dependently afterward. Five models simulate a decrease in the predictability of the neutral state, with HadCM3L showing the only significant change (-0.63, 15% relative to *control*; Fig. 7d). In summary, ENSO mean predictability decreases in *abrupt4xCO₂*, mainly caused by predictability changes in neutral and El Niño states. However, contradictory to our expectations, changes in predictability are neither as robust across models, nor as significant as changes in ENSO characteristics. Therefore, we test if ENSO characteristics-predictability relationship sustains in the warmer

climate.

### 3.3 Changes in ENSO characteristics-predictability relationship

#### 3.3.1 The relationship in the observations

In the observations, the characteristics-predictability relationship is principally significant for most of the analyzed characteristics for all states (Fig. 8a). The ENSO mean predictability increases when ENSO in general is stronger ($R^2$ = 62%) and less

red ($R^2$ = 42%). La Niña's predictability is higher when the events are stronger ($R^2$ = 39%), longer-lasting ($R^2$ = 21%) and the Niño3.4 distribution more skewed to favor the La Niña state ($R^2$ = 24%). Similarly, the neutral state is more predictable when the events are shorter ($R^2$ = 36%). As for El Niño's predictability, it only significantly relates to the intensity of El Niño events ($R^2$ = 24%). In summary, in the observations, ENSO predictability is affected by ENSO amplitude, which is attributed from the intensity of both El Niño and La Niña states.

#### 3.3.2 The relationship in *control*

In *control*, signs and significances of the characteristics-predictability relationship is model-dependent; the overall contribution of ENSO characteristics to ENSO predictability is lower than in the observations. For the mean, the relationship between ENSO predictability and ENSO amplitude is significant across four models (Fig. 8c). Looking at individual models, CNRMCM61





shows the largest $R^2$ value in relationships of the mean predictability to ENSO amplitude (18%) and frequency (14%), while
GISSE2R is the only model that shows significance in all three relationships but demonstrates extremely small $R^2$ values (1%
for amplitude, 1% for frequency, and 2% for skewness).

For active ENSO states, the relationships between El Niño's predictability and its characteristics are significant across at
least five models (Fig. 8c). Looking at individual models, MPIESM12 shows the largest $R^2$ values in relationships of the
predictability to El Niño's intensity (14%) and duration (19%), but the smallest $R^2$ value in the skewness-predictability rela-
tionship. Similarly, the characteristics-predictability relationships of the La Niña state are significant in at least five models.
CNRMCM61 shows the largest $R^2$ values in relationships of the predictability to La Niña's intensity (13%) and duration (17%).

For the neutral state, the relationship between its duration and predictability is significant for four models, while the sign of
the relationship is model-dependent. CESM104 shows the largest $R^2$ value (12%) in such a relationship, while CNRMCM61
shows the second largest $R^2$ (7%) but matches the sign of the relationship apparent in the observations (Fig. 8a).

In summary, the ENSO characteristics-predictability relationship is overall lower in the models in *control* than in the observa-
tions. In *control*, ENSO predictability is mainly affected by states' intensity and duration, counter to ENSO mean characteristics
which dominates in the observations.

### 3.3.3   The relationship in the changes

Moving from the relationship in the observations and *control*, we regress changes in ENSO predictability onto changes in
ENSO characteristics, for which the change is the difference between *control* and *abrupt4xCO₂*. Similar to the relationship
in *control*, the sign and significance of relationships of changes are model-dependent. The contribution of changes in ENSO
characteristics to predictability becomes more divergent across models but overall even lower than in *control*. The amplitude-
predictability relationship is significant across four models (Fig. 8d). Among models who show significant results, HadCM3L
has the largest $R^2$ value in the amplitude-predictability relationship (11%), and IPSLCM5A has the largest $R^2$ in the skewness-
predictability relationships (17%). However, models with the strongest relationships between changes in predictability and
changes in amplitude and skewness exhibit erroneous signs of the respective relationships compared to the observations.

For active ENSO states, on the one hand, the relationships between changes in El Niño's predictability and its characteristics
are significant across four models (Fig. 8d). Among models which show significant results, IPSLCM5A has the largest $R^2$ in the
relationships of changes in predictability to changes in El Niño's intensity (18%) and the skewness (23%); MPIESM12 has the
largest $R^2$ in the duration-predictability relationship (16%). On the other hand, the characteristics-predictability relationships
of changes in the La Niña state are significant in four of seven models (Fig. 8d). Among models which show significant results,
HadCM3L has the largest $R^2$ in the relationships of changes in predictability to changes in La Niña's intensity (27%) and
the skewness (40%), and IPSLCM5A has the largest $R^2$ in the duration-predictability relationship (35%). Four out of seven
models in *control* agree with the observations that El Niño and La Niña states typically end in spring (March – May; figure not
shown). In *abrupt4xCO₂*, active states end around 1 to 3 months earlier than in *control*, which shortens their decaying phase
and advances the transition to the neutral state. The persistence model fails to predict such a transition correctly, explaining the





sharp decrease in the persistence skills from February onward (Fig. 5) and leading to the decreased predictability (Fig. 6e and f).

For the neutral state, the relationship between changes in its duration and predictability is significant in five out of seven models, with four models showing a positive and one model showing a negative relationship (Fig. 8d). CNRMCM61 is the only model that shows a significant and negative duration-predictability relationship (8%), agreeing with the observations, while four models show a positive relationship with the mean $R^2$ value at 14%. For these four models, we find that the increased duration of the neutral state exercises two effects on its predictability. On the one hand, an increased duration of neutral states decreases the duration of active states. The decreased duration of active states fastens their transitions to the neutral state, which decreases the predictability of the neutral state, as its earlier occurrence cannot be predicted by the November initialization of the persistence model. On the other hand, the pronounced increase in the neutral state causes each neutral event to last on average 13 months longer in *abrupt4xCO₂* than in *control* (Fig. 6g). This cross-years duration increases the persistence of the neutral state and improves its predictability. Overall, because the predictability of the neutral state in these four models decreases (Fig. 7d), we deduce that the effect of early-occurrence of the neutral state during the spring overcompensates the effect of its cross-year duration on the predictability change.

We find that the relationships of ENSO states are more significant and have higher $R^2$ values than relationships of the mean. The characteristics we analyze are not independent, as they may offset each other's contributions or have two competing effects on the predictability change. Thus, we suspect that the mean characteristics (amplitude, frequency and skewness) may be too aggregated to be as clearly related to the mean predictability change as the states' characteristics (intensity and duration). Furthermore, given that ENSO is a highly non-linear process, it is likely that the three ENSO states respond to external forcings differently (e.g., Cai et al., 2015). This highlights the importance of decomposing different ENSO states from the mean ENSO behavior in order to study ENSO response to an external forcing.

In summary, $R^2$ values are relatively large in the observations, lower in *control*, more divergent and even lower in changes between *abrupt4xCO₂* and *control*. We suspect that it might be attributed to model parameterizations and tuning processes that models simulate ENSO events for wrong reasons (e.g., Bayr et al., 2019; Dommenget et al., 2014). However, the difference between the magnitudes of the feedback (Fig. 3) seems to be unrelated to magnitudes of the $R^2$ values between changes in ENSO characteristics and predictability.

## 4 Summary and conclusion

We study the equilibrated ENSO response to abruptly quadrupled $CO_2$ forcing in seven models from the LongRunMIP archive. We analyze changes of the whole time series of Niño3.4 index, as well as separate El Niño, La Niña and neutral states. Our results show that:

- ENSO mean characteristics significantly change in a warmer climate. The ENSO amplitude and the skewness of the Niño3.4 index distribution decreases, agreeing to the degree with findings in Callahan et al. (2021) and Kohyama et al.





(2018). The ENSO frequency decreases, which is different from the transient ENSO response analyzed in Berner et al. (2020).

– ENSO states' characteristics significantly change in a warmer climate. The decreased intensity of El Niño and La Niña states is different from the transient ENSO response shown in other studies (e.g., Cai et al., 2021).

From the robust response of ENSO characteristics under *abrupt4xCO₂* forcing, we would expect ENSO predictability to change similarly (e.g., Jin et al., 2008; Liu et al., 2022). We find that:

– ENSO predictability reduces for the mean and the three ENSO states in five out of seven models. However, the changes are small and statistically insignificant.

We find a relationship between ENSO characteristics and predictability in the observational record. We test whether models reproduce that relationship with and without forcing. We find that:

– In the observations, this relationship is strong. The mean ENSO characteristics and La Niña state's characteristics explain
more than 60% of the variance in ENSO predictability.

– In *control*, this relationship is weaker. Although the regression slopes significantly differ from zero, the ENSO characteristics only explain 20% of variance in ENSO predictability.

– In changes between *control* and *abrupt4xCO₂*, the regression slopes are significant. The decreased ENSO mean characteristics correlates to the decreased ENSO predictability, consistent with previous studies (e.g., DelSole et al., 2014;
Berner et al., 2020; Tang et al., 2008). The decreased intensity of active ENSO states correlates with the decreased predictability, consistent with results from Jin et al. (2008), and with the decreased duration, which reduces the persistence skills mainly in spring. While the slopes are significant, the explained variance of changes in ENSO predictability by ENSO characteristics is overall small, though some models show $R^2$ values being around 40%. This reduction in explained variances probably contributes to changes in ENSO predictability being insignificant.

Dommenget and Vijayeta (2019) studied the inconsistent change between ENSO characteristics and predictability under RCP8.5 scenarios relative to the historical simulation, which can be attributed to compensating effects of changes in the growth rate of the thermocline depth and sea surface temperature. While this might explain our results as well, we cannot verify this hypothesis, due to the lack of monthly ocean data in LongRunMIP. Aside from changes in ENSO characteristics studied here, other factors influence ENSO predictability: ENSO phase-locking (Jin and Kinter III, 2009), the Atlantic Niño
(Martín-Rey et al., 2015), and the Pacific Decadal Oscillation (Kumar et al., 2013). However, given our findings that changes in the persistence-assessed predictability are very small, we suspect changes in these factors barely contribute to predictability changes or offset each other. Instead of the persistence forecast, other dynamical prediction models could be used to analyze whether ENSO predictability in warmer climates is indeed unchanged, or, if a significant change is detected, whether its relationship to previously revealed ENSO characteristics sustains. In addition, ENSO predictability is also affected by model

development such as improved physical parameterizations, refined spatial and temporal resolutions, and reduced biases in air-sea interaction (e.g., Chen and Cane, 2008; Tang et al., 2018; Guilyardi et al., 2020; Lu et al., 2020; Fredriksen et al., 2020).

To conclude, if our set of simulations were representative of possible real-world changes, we do not find robust reasons to expect global warming to substantially alter six lead-month ENSO predictability. Yet, given the large range suggested by the different models, further analysis and deeper understanding of unforced control states and the expected changes under global 290 warming as well as their implications are necessary to be able to reduce the large uncertainties of the present assessment.

*Data availability.* https://mountainscholar.org/handle/10217/234545

*Author contributions.* Y. Z. and M. R. formulated research questions and goals. M. R. provided the LongRunMIP simulations. Y. Z., P. P. and G. B. conducted statistical analyses on the data. M. R. and J. B. supervised the research. Y. Z. produced figures and prepared the manuscript with contributions from all co-authors.

*Competing interests.* The authors declare that they have no conflict of interest.

*Acknowledgements.* The work of Y. Z., P. P. and J. B. was funded by the Deutsche Forschungsgemeinschaft (DFG, German Research Foundation) under Germany's Excellence Strategy – EXC 2037 'CLICCS - Climate, Climatic Change, and Society' – Project Number: 390683824, contribution to the Center for Earth System Research and Sustainability (CEN) of Universität Hamburg". This work used resources of the Deutsches Klimarechenzentrum (DKRZ). More information on LongRunMIP can be found under http://www.longrunmip. 300 org/.





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





**Table 1.** Overview of model simulations.

| Model name (short name) | Modelling center | Forcing levels | Simulation length (years) | Atmosphere resolution (latitude x longiture) |
|---|---|---|---|---|
| CCSM3 (**CCSM3**) | NCAR | control | 1530 | 3.8° x 3.8° |
| | | abrupt4x | 2120 | |
| CESM 1.0.4 (**CESM104**) | NCAR | control | 1320 | 1.4° x 1.4° |
| | | abrupt4x | 5900 | |
| CNRM-CM6-1 (**CNRMCM61**) | CNRS | control | 2000 | 1.4° x 1.4° |
| | | abrupt4x | 1850 | |
| GISS-E2-R (**GISSE2R**) | NASA | control | 5225 | 2.0° x 2.5° |
| | | abrupt4x | 1000 | |
| HadCM3L (**HadCM3L**) | Hadley Centre | control | 1000 | 2.5° x 3.8° |
| | | abrupt4x | 1000 | |
| IPSL-CM5A-LR (**IPSLCM5A**) | IPSL | control | 1000 | 1.9° x 3.8° |
| | | abrupt4x | 1000 | |
| MPI-ESM-1.2 (**MPIESM12**) | MPI | control | 1237 | 1.9° x 1.9° |
| | | abrupt4x | 1000 | |

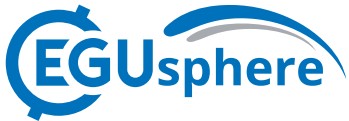

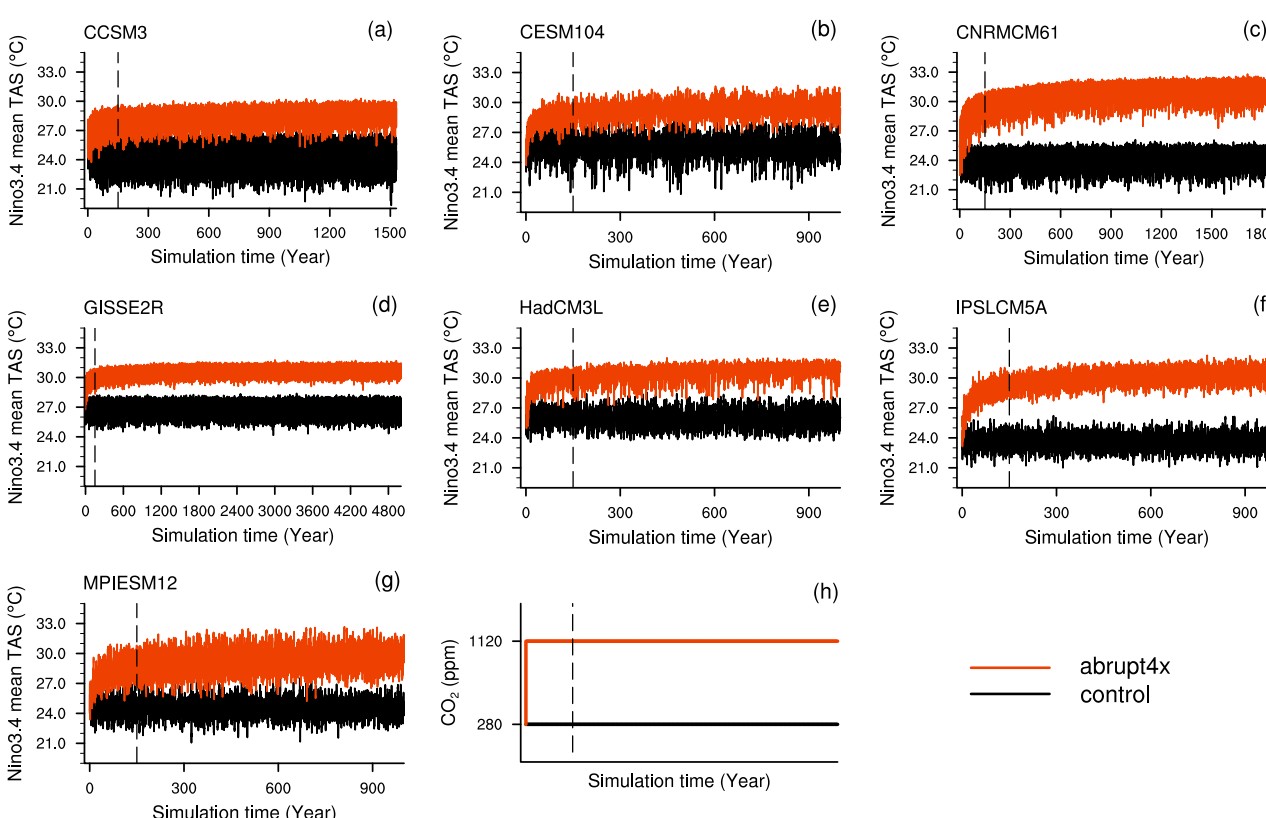

**Figure 1. (a – h)** Time series of Niño3.4 mean TAS and **(g)** CO$_2$ concentrations in control and abrupt4xCO$_2$ simulations. The dash lines mark the year 150, the period after which we assume a quasi-equilibrium.



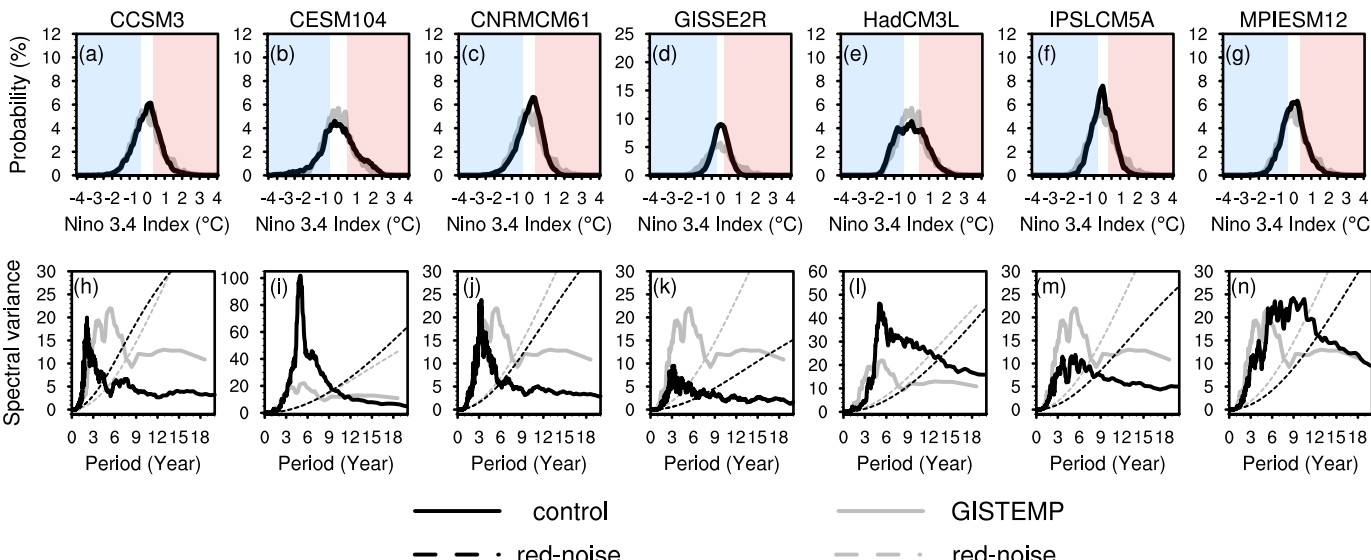

**Figure 2. (a – g)** Histograms and **(h – n)** power spectra of the Niño3.4 index in the observations and *control*. The blue and red shades in the upper panels represent La Niña and El Niño events, respectively. The dash lines in the bottom panels represent the 0.95 confidence bound of Markov red noise spectrum in the observations (grey) and control (black).



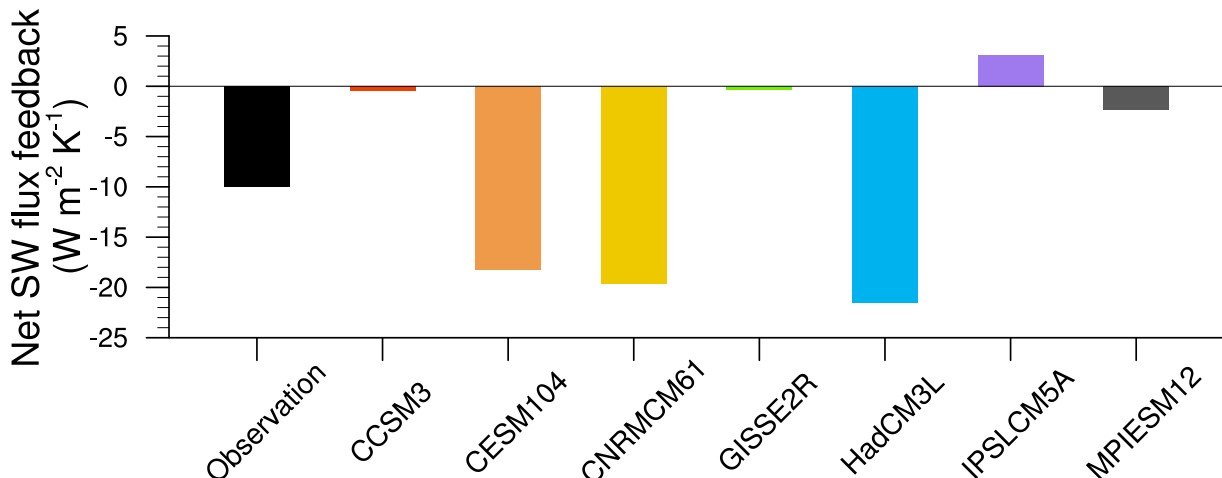

**Figure 3.** Net shortwave heat flux feedback in the observations and *control*.





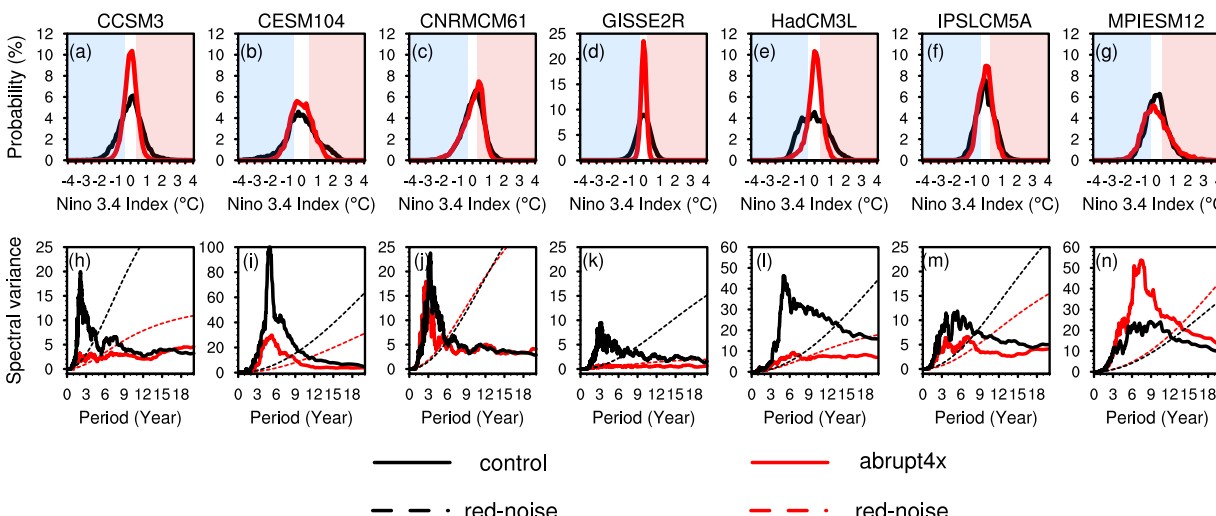

**Figure 4. (a – g)** Histograms and **(h – n)** power spectra of the Niño3.4 index in *control* and *abrupt4xCO₂*. The blue and red shades in the upper panels represent La Niña and El Niño events, respectively. The dash lines in the bottom panels represent the 0.95 confidence bound of Markov red noise spectrum in *control* and *abrupt4xCO₂*.





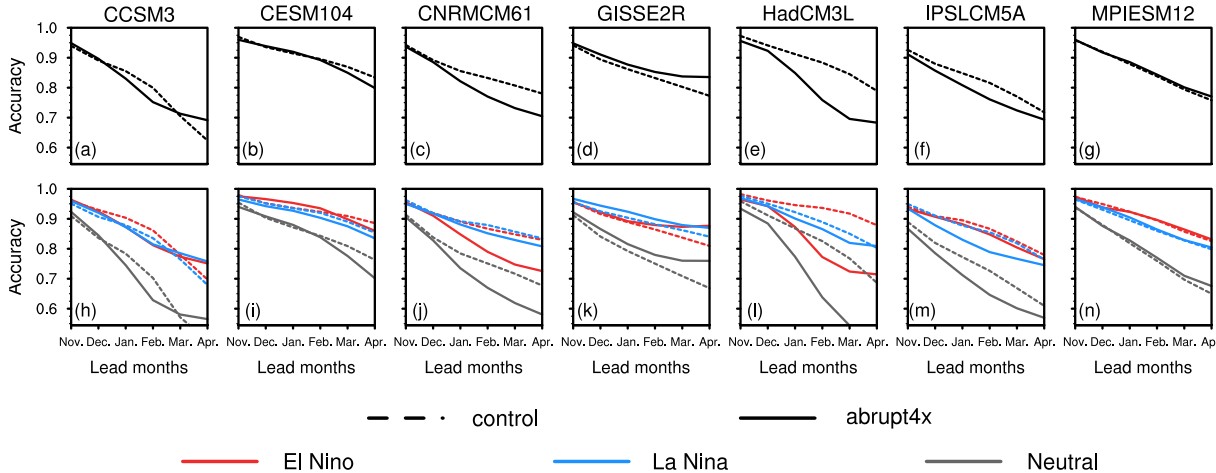

**Figure 5. (a – g)** Accuracy of persistence predictions of the mean ENSO state in *control* and *abrupt4xCO₂*. **(h – n)** Accuracy of persistence predictions of the El Niño, La Niña and neutral state in *control* and *abrupt4xCO₂*.





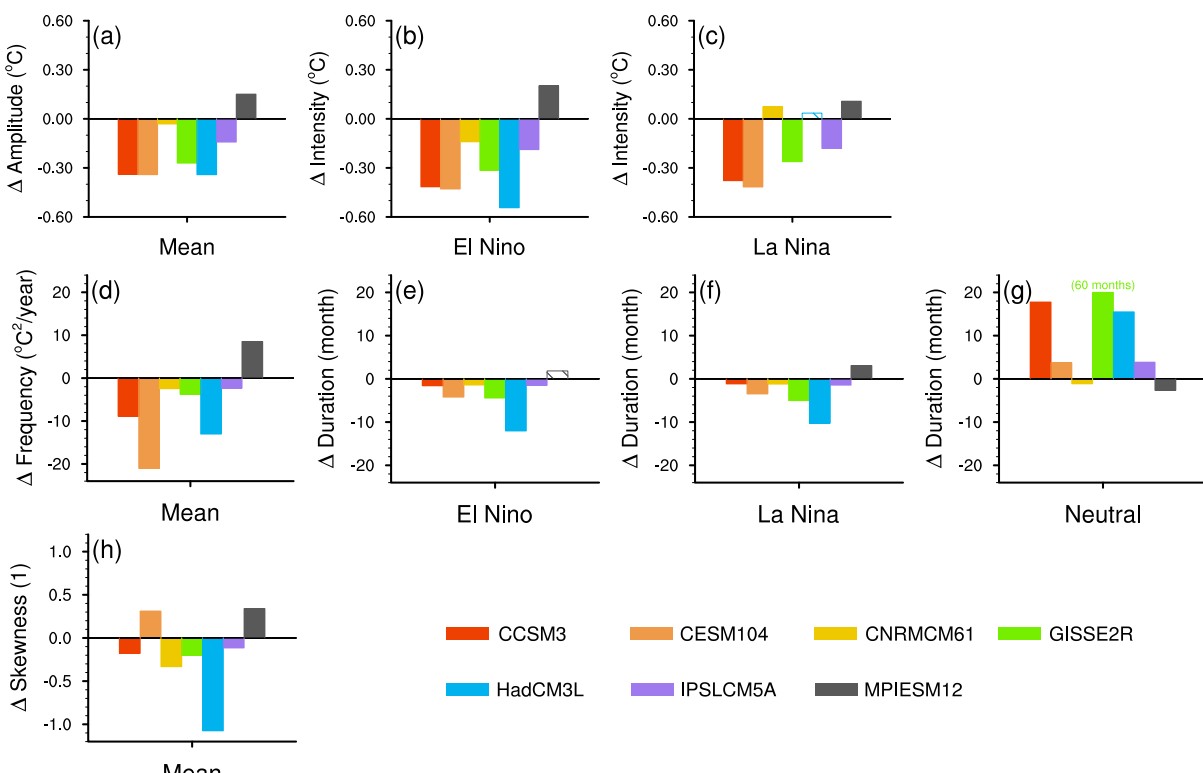

**Figure 6. (a – c)** Changes in *abrupt4xCO₂* relative to *control* in ENSO amplitude, and in intensity of El Niño and La Niña states. **(d – g)** Changes in *abrupt4xCO₂* relative to *control* in ENSO frequency, and in duration of El Niño, La Niña and neutral states. **(h)** Changes in the skewness of the Niño3.4 index distribution. Solid bars represent changes that are significant at the 5% level of a bootstrap test, hatched bars represent changes that are insignificant.





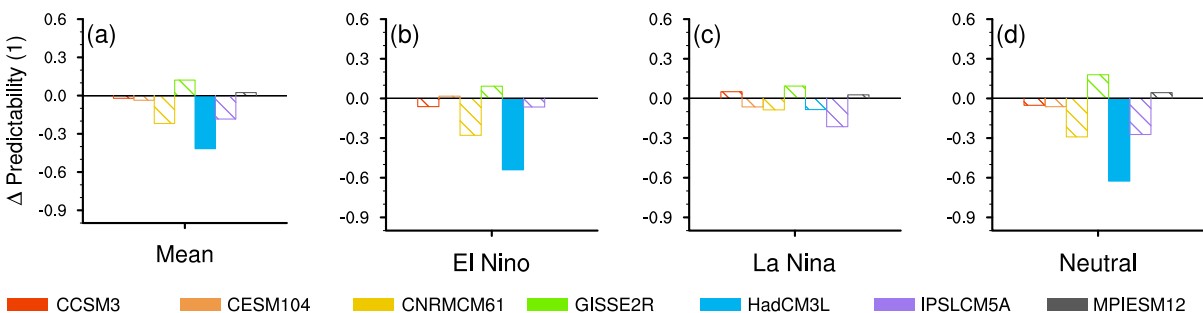

**Figure 7.** Changes in ENSO predictability in the mean state, in El Niño, La Niña and neutral states. Solid bars represent changes that are significant at the 5% level of a bootstrap test, while hatched bars represent changes that are insignificant.





**Figure 8. (a)** $R^2$ values in the observations, which are calculated from regression analyses between 10 ENSO metrics and predictability as shown in the label bar. **(b)** Example plot of the regression analysis that represents the uppermost bar in **(a)**. The small grey dots are 30-year chunks from the time series, and the line shows the regression slope. **(c)** Similar to **(a)** but for *control*. **(d)** Similar to **(a)** and **(c)** with regression analyses between changes in ENSO characteristics and predictability. The sign of bars represents the sign of regression slopes. The solid (hatched) bars represent regression slopes that significantly (insignificantly) differ from zero.