# Peer review of "ENSO predictability changes in an equilibrium warmer climate in LongRunMIP models"

_EGUsphere, 2022_

## Author Response (AR1)

++++++++++++++++++++++++++++++++

**Comment on egusphere-2022-89**

Anonymous Referee #1

++++++++++++++++++++++++++++++++

The study titled" Insignificant but robust decrease of ENSO predictability in an equilibrium warmer climate" investigates ENSO characteristics and predictability in a warmer climate. It is a novel work which furthers the understanding of ENSO predictability in the future. It can be accepted after the following comments are addressed:

Line 55: The authors should explain LongRunMIP in detail.

We apologize for the oversimplification. The explanation is added at Line 56, and the first paragraph of Section 2 becomes:

We analyze model outputs from the LongRunMIP archive, a large set of simulations from atmosphere-ocean general circulation models that last at least 1000 years (Rugenstein et al., 2019). This archive includes 15 models, each contains a pre-industrial control simulation of constant 1850-forcing and at least one forced simulation, ranging from instantaneous steps of two-, four-, eight-, or sixteen-time $CO_2$ to realistic Representative Concentration Pathway (RCP) scenarios with constant forcing beyond year 2300. We select seven models from the archive with abrupt quadrupling $CO_2$ simulations (hereinafter referred to as *abrupt4xCO2*) and compare them with their pre-industrial control simulations (hereinafter referred to as *control*). This forcing level is chosen because it shows a better signal-to-noise ratio than the doubling of $CO_2$. The spatial resolution of these seven models ranges from 3.75° in both the atmosphere and ocean to 1.4° in the atmosphere and 0.5° in the ocean. Though models with different resolutions might show different ENSO dynamics, our results on ENSO response are robust among the models. We treat the period between year 150 and the end of the simulation as……

Line 250: The authors claim that "ENSO mean characteristics significantly change in a warmer climate.". Can the authors verify this with the paleoclimate research on ENSO when the Earth's temperature was warmer than the present day? Support from proxy records would justify the analysis considering the biases in the models.

We add it to the Line 256 and the second paragraph in Section 4 becomes:

The weakening in ENSO characteristics can be seen in both ultra-high-resolution models and proxy records. Wengel et al. (2022) showed a decreased ENSO amplitude in a model with 0.25° atmospheric resolution, which is both qualitatively and quantitatively consistent with our results (compare their Fig. 4a with our Fig.6a). White and Ravelo (2020) showed a reduced El Niño amplitude during early Pliocene, during which the climate resembles closely the one projected in RCP4.5 scenario (Burke et al., 2018).

From the robust response of ENSO characteristics under *abrupt4xCO2* forcing, we would expect ….

The authors discuss about the ENSO predictability, however, there is no mention of forcings which can make the ENSO more predictable such as volcanoes. Recent studies such as **Khodri et al 2017, Singh et al 2020** etc have shown that strong volcanic eruptions can trigger El Ninos.

**Khodri, M., Izumo, T., Vialard, J., Janicot, S., Cassou, C., Lengaigne, M., Mignot, J., Gastineau, G., Guilyardi, E., Lebas, N. and Robock, A., 2017. Tropical explosive volcanic eruptions can trigger El Niño by cooling tropical Africa. Nature communications, 8(1), pp.1-13.**

**Singh, M., Krishnan, R., Goswami, B., Choudhury, A.D., Swapna, P., Vellore, R., Prajeesh, A.G., Sandeep, N., Venkataraman, C., Donner, R.V. and Marwan, N., 2020. Fingerprint of volcanic forcing on the ENSO–Indian monsoon coupling. Science advances, 6(38), p.eaba8164.**

The suggested paper and a short discussion of the relevance of volcanoes now included at Line 280: Aside from changes in ENSO characteristics studied here, other factors influence ENSO predictability: ENSO phase-locking (e.g. Jin and Kinter III, 2009), the Atlantic Niño (e.g. Martín-Rey et al., 2015), the Pacific Decadal Oscillation (e.g. Kumar et al., 2013), and volcanic eruptions (e.g. Khodri et al., 2017; Singh et al., 2020).

++++++++++++++++++++++++++++++
**Comment on egusphere-2022-89**
Anonymous Referee #2
++++++++++++++++++++++++++++++

In this manuscript the authors have examined the predictability of ENSO in coupled ocean-atmosphere models whose simualtions are more than 1000 years long. Many of the conclusions about the changes in ENSO charateristics is wellknown based on previous work in this area. The main advantage of this work is their ability to look at a much longer simualtion and hence differentiate between transinet and equlibirum response. The authors have not discussed the role of spatial resolution on the simualtion of ENSO. This is important since a recent work by Wengel et al, Nature Climate Change ,2022 "Future high-resolution El Niño/Southern Oscillation dynamics" has claimed that the resolution of mesoscale processes is important to simulate the robust weakening of ENSO with global warming. Hence there is a need to add some discussion about this paper and convince the readers that the results reported in the manuscript are not sensitive to spatial resolution

We add to both Line 59 and Line 256, now the first paragraph in Section 2 and the second paragraph in Section 4 become:

We analyze model outputs from the LongRunMIP archive, a large set of simulations from atmosphere-ocean general circulation models that last at least 1000 years (Rugenstein et al., 2019). This archive includes 15 models, each contains a pre-industrial control simulation of constant 1850-forcing and at least one forced simulation, ranging from instantaneous steps of two-, four-, eight-, or sixteen-time $CO_2$ to realistic Representative Concentration Pathway (RCP) scenarios with constant forcing beyond year 2300. We select seven models from the archive with abrupt quadrupling $CO_2$ simulations (hereinafter referred to as *abrupt4xCO2*) and compare them with their pre-industrial control simulations (hereinafter referred to as *control*). This forcing level is chosen because it shows a better signal-to-noise ratio than the doubling of $CO_2$. The spatial resolution of these seven models ranges from 3.75° in both the atmosphere and ocean to 1.4° in the atmosphere and 0.8° in the ocean. Though models with different resolutions might show different ENSO dynamics, our results on ENSO response are robust among the models. We treat the period between

year 150 and the end of the simulation as……

The weakening in ENSO characteristics can be seen in both ultra-high-resolution models and proxy records. Wengel et al. (2022) showed a decreased ENSO amplitude in a model with 0.25° atmospheric resolution, which is both qualitatively and quantitatively consistent with our results (compare their Fig. 4a with our Fig.6a). White and Ravelo (2020) showed a reduced El Niño amplitude during early Pliocene, during which the climate resembles closely the one projected in RCP4.5 scenario (Burke et al., 2018).
From the robust response of ENSO characteristics under *abrupt4xCO₂* forcing, we would expect ….